# Theoretical Evaluation of Graphene Membrane Performance for Hydrogen Separation Using Molecular Dynamic Simulation

**DOI:** 10.3390/membranes9090110

**Published:** 2019-08-27

**Authors:** Mahdi Nouri, Kamran Ghasemzadeh, Adolfo Iulianelli

**Affiliations:** 1Faculty of Chemical Engineering, Urmia University of Technology, Urmia 57155-419, Iran; 2Institute on Membrane Technology of the Italian National Research Council (CNR-ITM), via P. Bucci 17/C, 87036 Rende (CS), Italy

**Keywords:** graphene membrane, molecular dynamic simulation, hydrogen separation

## Abstract

The main purposes of this study are to evaluate the performance of graphene membranes in the separation/purification of hydrogen from nitrogen from a theoretical point of view using the molecular dynamic (MD) simulation method, and to present details about molecular mechanisms of selective gas diffusion through nanoscale pores of graphene membranes at the simulated set conditions. On the other hand, permeance and perm-selectivity are two significant parameters of such a membrane that can be controlled by several variables such as pressure gradient, pore density, pore layer angles etc. Hence, in this work, the hydrogen and nitrogen permeating fluxes as well as the H_2_/N_2_ ideal perm-selectivity are investigated from a theoretical point of view in a two-layer nanoporous graphene (NPG) membrane through classical MD simulations, wherein the effects of pressure gradient, pore density, and pore angle on the NPG membrane performance are evaluated and discussed. Simulation outcomes suggest that hydrogen and nitrogen permeating fluxes increase as a consequence of an increment of pressure gradient across the membrane and pore density.

## 1. Introduction

The increasing demand for highly efficient operations resulted in increased global willingness to embrace membrane technology as a potential long-term solution to mitigate the emission of gases that contribute to global warming. Indeed, with global energy consumption predicted to nearly double by 2050 and the increasingly urgent environmental and economic pressures our present fossil fuel reserves are under, humans must unambiguously overcome many scientific and technological hurdles that exist between the present state of membrane production, utilization, and potential applications. Nowadays, worldwide environmental concerns are triggering the search for environmentally friendly materials to be used in industry. On the other hand, gas separation processes such as hydrogen purification are very important, particularly in the framework of hydrogen economy development [1,2].

In the last three decades, membrane hydrogen separation also became popular at a commercial scale, owing to low energy costs (in case of polymeric membranes utilization), absence of chemical processing, flexible structures, etc. The most adopted membrane material for hydrogen separation/purification was palladium [3,4], which became the dominant option in case of high-grade hydrogen separation over other inorganic membranes based on silica [5,6,7] and zeolite [8,9]. Nevertheless, to maximize membrane performance, one should consider the trade-off between hydrogen permeability and selectivity by optimizing the material structures and morphology. The increase of pressure-drop across the membrane and the lower hydrogen permeabilities as a consequence of thicker membranes utilization suggests that ultrathin membranes should be preferred. However, to maintain their structural integrity under operation, remarkable mechanical resistance to the pressure load should be ensured. In recent years, graphene membranes were considered to be excellent solutions to the aforementioned issues in various applications due to the nature of their single-atom thickness, excellent mechanical resistance, and chemical stability [10]. As a result, ultrathin graphene membranes found a promising application in the field of hydrogen purification. Unfortunately, to date, there are not a large number of experimental studies dealing with performance comparison of graphene membranes in the field of hydrogen separation [11]. Therefore, taking into account the high cost of experimental campaigns, modeling, and simulation of hydrogen separation/purification processes [12,13,14,15,16], such as the molecular dynamic (MD) method [17,18,19,20], could be useful to achieve a better understanding of the effects of several parameters for the design and performance of graphene membranes and also specific features and constraints like the need to obtain as pure high-purity hydrogen as possible. To this purpose, the MD tool is a feasible method to simulate detailed gas flow characteristics of a membrane system. Indeed, the MD approach can be used for virtual analyses of membrane separators based on a statistical methodology [17]. In particular, the local variations of the fluid and mass transport properties can be observed by simple models to design the graphene membrane cell. To the best of our knowledge, only a few MD studies deal with the graphene membrane performance in the framework of hydrogen separation processes. Therefore, in the present work, as a first approach, a two-layer graphene membrane cell is theoretically studied by applying the MD method to investigate the effects of the most important design parameters such as pore density and pore angles on graphene membrane performance in terms of hydrogen permeating flux and H_2_/N_2_ ideal perm-selectivity, during various pressure gradients.

## 2. MD Simulation Methodology

### 2.1. Preparation of Virtual Membrane Cell

A schematic of a nanoporous graphene (NPG) model is shown in Figure 1a, which is represented by a square graphene membrane with an area of 4.5 × 4.5 nm². The main purpose of this model consists in the evaluation of pore angle effects on graphene membrane performance. A double NPG sheet is placed at the center of the box in the *xy* plane, and the inter space of graphene layer was set equal to 1 nm. Considering the study of Huailiang et al. [21], periodic boundary conditions were adopted for all three dimensions. Hence, by considering these conditions, gas and vacuum phases are separated by a fixed non-porous graphene sheet. The pores are formed by removing three neighboring benzene rings in each graphene sheet, which are placed at four different angles (0°, 15.83°, 30.45°, and 40.36°) with respect to the other sheets, as illustrated in Figure 1b,e. It should be considered that, for this case, the pore density of each graphene sheet was calculated at about 3%.

More in detail, to evaluate the pore density effect on graphene membrane performance, the second and third cases of NPG models are defined as illustrated in Figure 2a,b, respectively. The models show the same structure as for the first model (Figure 1). Only in these cases, the pore angle was considered equal to 15.83°, while pore density was changed from 5% to 8% in the second and third models, respectively. On the other hand, in all defined model boxes, an equal number of molecules (100) for nitrogen and hydrogen were considered for the simulations.

### 2.2. Solution Procedure

The molecular displacement was traced via the integration algorithm of Velocity-Verlet based on Newton’s second law. Classical MD simulations are performed using the Large-scale Atomic/Molecular Massively Parallel Simulator (LAMMPS) (26 Jan 2017 version, CA, USA) [18] in a canonical NVT (constant number of atoms, volume and temperature) ensemble. All simulations are performed for a duration of 15 ns with a time step of 0.25 fs. Calculated parameters however are stored following every 8000 time steps to analyze the molecular permeation event and other quantities. The Nose-Hoover thermostat is employed to keep the system at a temperature of about 300 K and each model is simulated with four different initial pressures (1, 3, 5 and 10 bar) in the gas phase. The gas pressure is calculated by the ideal gas equation. The Lennard–Jones (LJ) potential can be defined as:(1)$$U_{\mathrm{ij}}=4\varepsilon [(\frac{\delta }{r}{)}^{12}-{\left(\frac{\delta }{r}\right)}^{6}]$$
where *ε* is depth of the potential well, σ is finite distance at which the inter-particle potential is zero and r is distance between the particles. The presented parameters in Table 1 are used to define behavior the interatomic interactions, while the harmonic potential is given by:E = K(r − r_0_)^2^(2)
where K (energy/distance^2^) is the bond coefficient and r_0_ (distance) is the equilibrium bond distance calculated from Gauss View version 5.0 (United states Pennsylvania), which is used for modeling the internal N–N, H–H and C–C bonds with parameters listed in Table 2.

## 3. Results and Discussions

As mentioned before, a run series of simulations based on MD method was performed to evaluate the effects of imperative parameters (pore angles and pore density) on NPG membrane performance in terms of gas permeating flux and H_2_/N_2_ ideal perm-selectivity, as presented in the following section.

### 3.1. Evaluation of Pore Angle Effects on NPG Membrane Performance

Figure 3a indicates that hydrogen and nitrogen permeating fluxes decrease by increasing the pore angle at 10 bar of pressure gradient. However, the decreasing trend of the hydrogen permeating flux is observable only with pore angles higher than 15, while for nitrogen it is observable in pore angles range from 0 to 40. This is probably due to lower interaction effects of hydrogen molecules over those of nitrogen at low pore angles. On the contrary, the H_2_/N_2_ ideal perm-selectivity of the NPG membrane increases by increasing the pore angle (see Figure 3b). Indeed, this result denotes the enhancement of NPG membrane tortuosity as a consequence of increasing pore angle.

Hence, the tortuosity of pore canals in graphene membranes is favorable for improving the membrane selectivity. Therefore, it can be concluded that pore tortuosity of graphene membrane can be increased by the enhancement of graphene coating layers. However, to the best of our knowledge, there were no works about theoretical evaluation of this phenomenon until now. In fact, this achievement can be studied by experimental methods to obtain more certain results.

### 3.2. Evaluation of Pore Density Effects on NPG Membrane Performance

To evaluate the pore density effect on graphene membrane performance, considering the compromise of high H_2_ permeating flux and as high as possible H_2_/N_2_ ideal perm-selectivity as a function of pore angle, a value equal to 15.84 was set for the former. Being that the pressure gradient is one of the most significative parameters representing the driving force in membrane processes, pore density effect was evaluated at various pressure gradients. The hydrogen and nitrogen kinetic diameters are 2.89 Å and 3.64 Å [26,27], respectively, and the pore size is 4.48 Å. Hence, pore size is large enough for hydrogen to pass easily through the graphene membrane pores. The MD simulation results show that hydrogen can permeate at low pressure gradients. On the contrary, due to the larger kinetic diameter of nitrogen and larger interaction force between the latter and the NPG membrane, a higher driving force is necessary to pass through the pore for nitrogen. Therefore, the latter cannot permeate easily in low pressure gradients during 15 ns of simulation run time and, consequently, high H_2_/N_2_ ideal perm-selectivity can be reached. Furthermore, as theoretically predicted, by increasing pore density with the same pore size can cause component permeance to rise. Hence, as shown in Figure 4a hydrogen permeating flux upturns by increasing the pore density at an equal pressure gradient. Nevertheless, at a pore density and pressure gradient higher than or equal to 5% and 5 bar, respectively, the hydrogen permeating flux variation is not sensible. This result is probably related to batch conditions in the MD system. Figure 4b also indicates that nitrogen permeating flux increases continuously by raising pore density at high pressure gradients. This is probably due to a low interaction effect between hydrogen and nitrogen molecules. Therefore, it is worth noting that an increase of pore density involves higher permeating fluxes. However, it can negatively affect the perm-selectivity of NPG membrane during a separation process. On the other hand, Figure 4c illustrates that, in the theoretical modeling of NPG membranes with 8% of pore density and 10 bar of pressure gradient, in the first run time of 5,000,000 fs, 47 hydrogen molecules crossed the NPG membrane, equivalent to around 50% of the total number of hydrogen molecules. In addition, a notable variation in the number of crossed molecules during further run time steps (10,000,000 fs) was not observed. While the number of nitrogen molecules passing through the NPG membrane during first run time (5,000,000 fs) is very low, it was increased in the successive run time steps (10,000,000 fs). This may be explained with a reduced partial pressure of hydrogen in the feed side, determining that nitrogen molecules can pass through the NPG membrane (being most of hydrogen molecules already permeated), especially at high pressures. Hence, Figure 4d shows the H_2_/N_2_ ideal perm-selectivity at various pore densities and pressure gradients, illustrating how in low pressure and pore density region, high perm-selectivity can be reached. However, owing to an increase of nitrogen permeating flux at high pressures, the H_2_/N_2_ ideal perm-selectivity of the NPG membrane is depleted by increasing pore density.

A further comparison between diffusion of hydrogen and nitrogen molecules is shown in Figure 5, where the interaction energy of hydrogen and nitrogen molecules with the graphene layers is observed. As expected, higher repulsion power is obtained between nitrogen molecules and the graphene layer surface, while hydrogen molecules show two times less repulsion power with the graphene layer. Indeed, this result is a typical validation for simulation results illustrated in Figure 3 and Figure 4.

## 4. Conclusions

In this work, a MD simulation study has been presented for hydrogen separation processing by a two-layer NPG membrane, which elucidates the molecular mechanisms of selective gas diffusion through nanoscale pores of a graphene membrane. In particular, the simulation results show that, by increasing the pressure gradient, nitrogen and hydrogen permeating fluxes improve, even though the H_2_/N_2_ ideal perm-selectivity decreases as a function of pore densities. This was explained with the high competition effect between nitrogen and hydrogen molecules during the permeation process. On the other hand, this theoretical investigation pointed out how, at a set pressure gradient, by increasing the pore density of NPG membrane, the permeating fluxes of nitrogen and hydrogen increase, while H_2_/N_2_ perm-selectivity decreases, especially at high pressure gradients. This happens since higher pore densities mean higher membrane porosity, which determines an enhancement of the membrane permeating flux. Pore angle variation of graphene membrane layers was another parameter evaluated in this MD study. It was simulated that, at 10 bar of pressure gradient, hydrogen and nitrogen permeating fluxes decrease by increasing the pore angle, although the H_2_/N_2_ perm-selectivity increases. Hence, this theoretical study indicated the relevant role of managing and synthesizing graphene membrane layers as a critical parameter to improve NPG membrane performance.

## Figures and Tables

**Figure 1 membranes-09-00110-f001:**
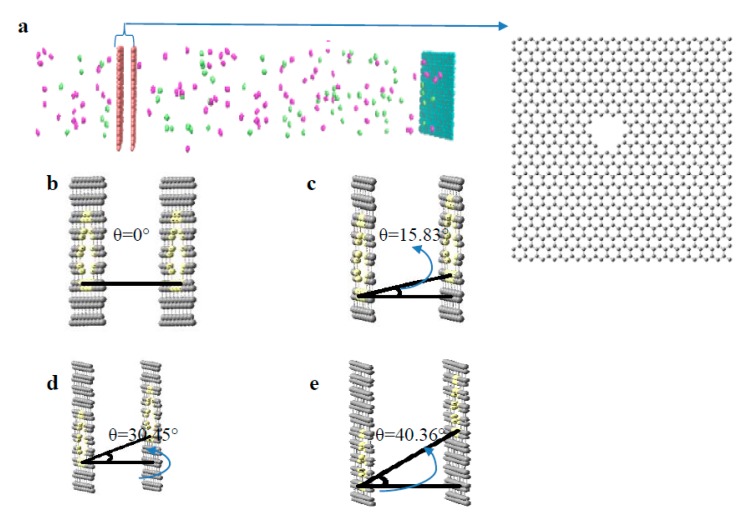
Molecular dynamic (MD) simulation classification and nano pores angle assemblies; (**a**) simulation organization and structure of each one of nanoporous graphene (NPG) layer with pore density 3%, (**b**–**e**) pores angles between two graphene layers.

**Figure 2 membranes-09-00110-f002:**
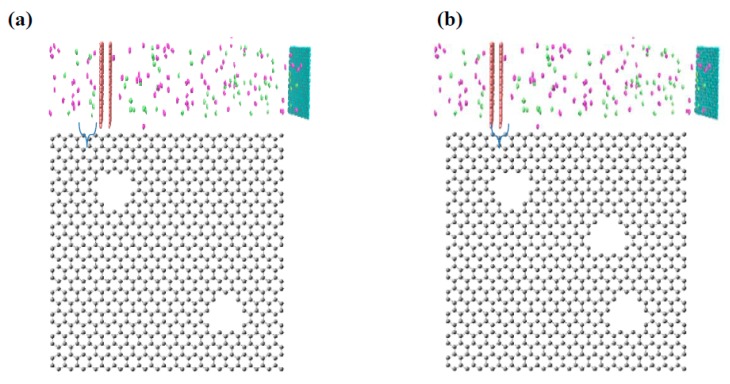
(**a**) Simulation organization and structure of each one of NPG layer with pore density 5%; (**b**) simulation organization and structure of each one of NPG layer with pore density 8%.

**Figure 3 membranes-09-00110-f003:**
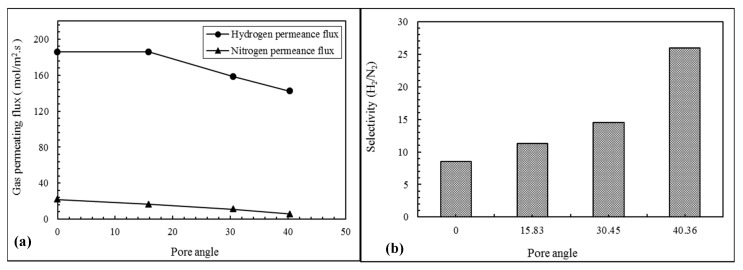
(**a**) Permeate flux of the hydrogen and nitrogen permeate fluxes (mol/m^2^·s) for various pore angles; (**b**) selectivity H2 /N2 for various pore angles.

**Figure 4 membranes-09-00110-f004:**
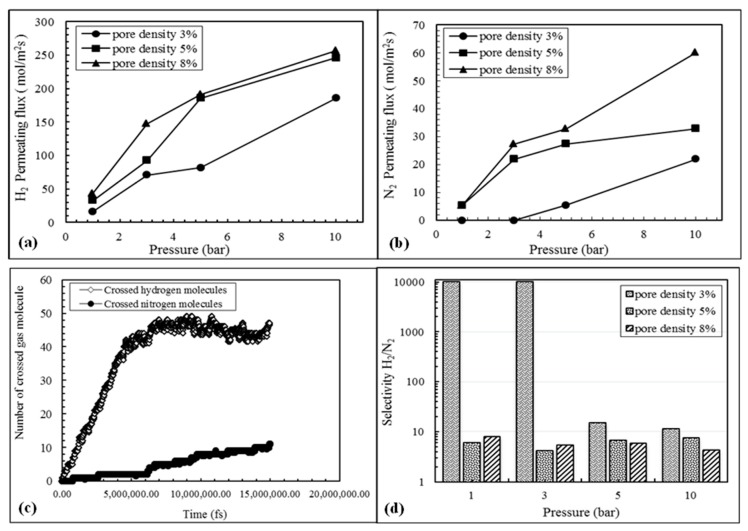
(**a**) Permeate flux of the hydrogen (mol/m^2^s), (**b**) permeate flux of nitrogen (mol/m^2^s) for various pore densities, (**c**) number of crossed gas molecule for pore density of 5%, (**d**) selectivity of H_2_/N_2_.

**Figure 5 membranes-09-00110-f005:**
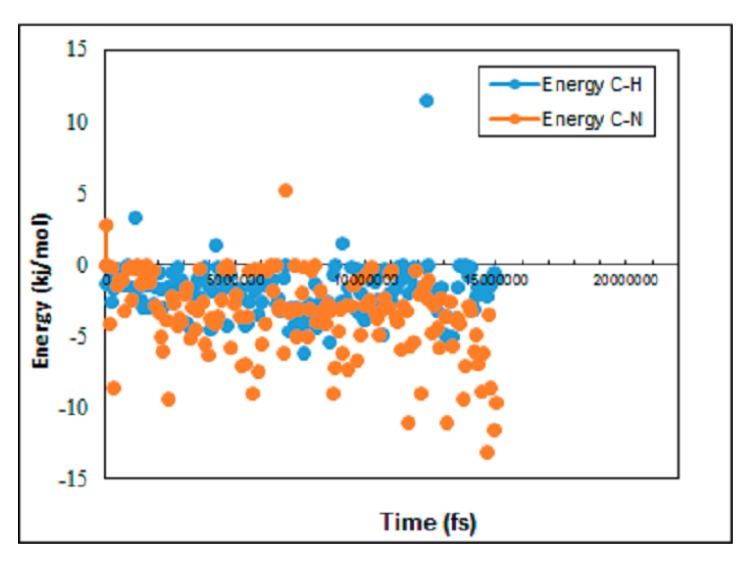
Energy profile for hydrogen and nitrogen molecules with the graphene layer versus time for pore density of 8% and pressure gradient 10 bar.

**Table 1 membranes-09-00110-t001:** Lennard-Jones interaction parameter.

Atom	(nm)	ɛ/kb (K)
Hydrogen [19]	0.2960	34.2
Nitrogen [20]	0.3798	71.4
Carbon [22]	0.3400	28.0

**Table 2 membranes-09-00110-t002:** Interatomic interaction parameter.

Bond	Energy (kcal/mol)	Distance (Å)
H-H [23]	104.2	0.74
N-N [24]	225.96	1.0975
C-C [25]	88.277	1.4177

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
