# Peer review of "Theoretical Evaluation of Graphene Membrane Performance for Hydrogen Separation Using Molecular Dynamic Simulation"

_membranes, 2019, doi:10.3390/membranes9090110_

Round 1
Reviewer 1 Report
Article entitled "Theoretical evaluation of graphene membrane performance for hydrogen separation using molecular dynamic simulation" describes from theoretical point of view the performance of graphene membranes in the separation of hydrogen from nitrogen using the molecular dynamic (MD) simulation method.
Authors show straight forward approach that is roughly describing the diffusion process of H2/N2 gas through defects in double graphene layer. Proposed approach does not include multiple additional effects e.g. graphene curvature, magnetic moments created on hole defects, different defect shapes.
Please address the following points in the revised version:
What is the used stacking of graphene layers is A-A, A-B type and distance between them?
Authors do not describe how to compare the theoretical results with real life samples. The closest structure to this one (as I know) is reduced graphene oxide paper where stacking is random flake size is random and number of layers is much larger. Could you extrapolate the results on real life samples, because in my opinion the number of additional configurations of flakes in larger (real) samples would change the outcome largely.
How many gas molecules were used in simulation and how many carbon atoms were in the graphene layer?
The orientation of holes in between layers was the same or were they rotated in the graphene plane. Additional effect which was not accounted for.
What is the difference between σ distance between NN HH CC in table 1 and the hydrogen and nitrogen kinetic diameters mentioned later in text. The values are very similar but different.
The value pore size is 4.48 A should be shown in the Fig 1 for clarification.
“Periodic boundary conditions are adopted for all dimensions.” Not all but for two, if I understand it correctly. If not please clarify.
Text corrections
“More in detail, To evaluate the pore density effect on graphene membrane performance, the second (Fig 1 b) and third (Fig 1c) cases of NPG models are defined as illustrated…” please add Figs for clarification.
Figure 1 a, b, c, d, e on the image tabulation and space marks visible and description is partially bold partially normal
“...depth of the potential well, σ is Finite distance at which the...” capital letter and potential well underlined and different font type
Fig 4 c scale and description are one over the other
Fig 4c and d missing legend description
In references line 202, 203 and 226 names are underlined and in some cases 223, 224, 244 they are hyperlinks.
Author Response
Responses to Reviewer comments:
Journal: Membranes
Title: Theoretical evaluation of graphene membrane performance for hydrogen separation using molecular dynamic simulation
Authors: M. Nouri, K. Ghasemzadeh, A. Iulianelli
Dear Dr. Li
Assistant Editor
We are very pleased to have been given the opportunity to revise our manuscript. We would like to thank you for your attention and also thank the referees for their useful comments.
We have considered all referee comments carefully and either made appropriate changes in the revised manuscript or gave suitable explanations. Our replies to referee’s comments are listed at the bottom of this letter.
We hope that these revisions improve the paper such that you and the respected reviewers now deem it worthy of publication in your journal. All the changes that we made in the manuscript are highlighted in yellow color in the revised manuscript.
With my all best regards
Corresponding Author
Dr. Kamran Ghasemzadeh
Associate Professor of Chemical Engineering
Director of UUT Research Center
Urmia University of Technology, Urmia, Iran
Tel: +989143888435
Email: [email protected]
Reviewer 2 Report
This manuscript evaluated the performance of double-layered graphene membranes in the separation/purification of hydrogen from nitrogen in a theoretical way. The manuscript was well written. However, after carefully reading and comparing with other work. I still doubt about the data source, i.e., where were the results analyzed from. Therefore, I suggest to publish after major revision.
1. The simulation images in Figure 1a and 2 are the same. I wonder why they are the same. They should be different under different condition.
2. In Fig. 3a and 4c, the label of N2 flux was not marked.
3. How were the pore angle effects performed and evaluated? Please provide the theoretical details, not simple description.
4. How was the selectivity obtained? Please compare with your work and others’ work in the section 3.1. Much more description and comparison should be included in section 3.1.
5. For Fig. 4d, there should be differences in the columns, either in color or in texture. Please mark the pore density 8% in the label.
6. As there are lots of work regarding “graphene membrane performance for hydrogen separation using molecular dynamic simulation”. Please refer to them and comparing with your work in section 3.2. Only discussion of the results in this paper is far from enough.
7. All the data points in the figures should be collected from at least three experiments. There should be error bars.
8. As this is a simulated experiment, the representative energy profiles should be provided.
Author Response

(The authors gave the same response as above.)

Round 2
Reviewer 2 Report
Dear Editor,
This corrected version can be accepted as publication.